# FAVAE-Effective Frequency Aware Latent Tokenizer

## Abstract

*Latent generative models have shown remarkable progress in high-fidelity image synthesis, typically using a two-stage training process that involves compressing images into latent embeddings via learned tokenizers in the first stage. The quality of generation strongly depends on how expressive and well-optimized these latent embeddings are. While various methods have been proposed to learn effective latent representations, the reconstructed images often lack realism, particularly in textured regions with sharp transitions, due to loss of fine details governed by high frequencies. We conduct a detailed frequency decomposition of existing state-of-the-art (SOTA) latent tokenizers and show that conventional objectives inherently prioritize low-frequency reconstruction, often at the expense of high-frequency fidelity. Our analysis reveals these latent tokenizers exhibit a bias toward low-frequency information, when jointly optimized, leading to over-smoothed outputs and visual artifacts that diminish perceptual quality. To address this, we propose a wavelet-based, frequency-aware variational autoencoder (FA-VAE) framework that explicitly decouples the optimization of low- and high-frequency components. This decoupling enables improved reconstruction of fine textures while preserving global structure. Our approach bridges the fidelity gap in current latent tokenizers and emphasizes the importance of frequency-aware optimization for realistic image representation, with broader implications for applications in content creation, neural rendering, and medical imaging.*

## 1 Introduction

Latent generative modeling [1, 2, 3, 4] has emerged as a cornerstone of modern content creation, with recent advances demonstrating remarkable capabilities in synthesizing high-fidelity visual content. These models typically operate in compressed latent spaces learned via autoencoders such as VAEs [5, 6], where generation quality is directly influenced by the expressiveness of these latent embeddings. While increasing latent dimensionality can enhance representational power, this often leads to diminishing visual outputs and increased computational cost. VAVAE [7] addresses this by aligning the latent space with pre-trained vision foundation models to further improve convergence and generative realism. However, despite such improvements, generated outputs or even the reconstructed outputs often lack sharp textures and fine details, particularly in regions dominated by the high-frequency information such as text on images. This limits perceptual realism and results in overly smooth outputs. Prior works [8, 9, 10] have explored architectural and spectral enhancements to inject high-frequency signals, but a systematic frequency-level analysis of how these current latent tokenizers influence reconstruction quality, especially in differentiating low- vs. high-frequency information remains absent.

In this work, we conduct a comprehensive frequency-based analysis of the reconstruction behavior of state-of-the-art latent tokenizers used in generative pipelines [7, 6, 11, 12, 13], focusing on latent diffusion and autoregressive models, which represent the current forefront of high-quality

Submitted to 39th Conference on Neural Information Processing Systems (NeurIPS 2025). Do not distribute.

visual generation. These models typically follow a two-stage pipeline: (1) learning quantized or non-quantized latent embeddings using a latent tokenizer like VAE or its variant, and (2) training a probabilistic generative model in the latent embedding space. Our analysis reveals a consistent bias during the first stage of learning latent embeddings: while low-frequency components are on par well reconstructed, high-frequency signals such as textures are poorly preserved. This low frequency bias over high frequencies in optimization contributes significantly to perceptual degradation.

To address this fidelity gap, we propose a novel **Frequency-aware VAE (FA-VAE)** latent tokenizer that explicitly decouples and separately optimizes low- and high-frequency components using wavelet decomposition. Our method learns distinct latent embeddings for low-high frequency subbands and later fuses them into a unified latent space representation. This simple design allows for better preservation of both global structure and fine details. Figure 1 presents a visual comparison between our method and the recent state-of-the-art VAVAE, highlighting improved preservation of fine details and sharp structures.

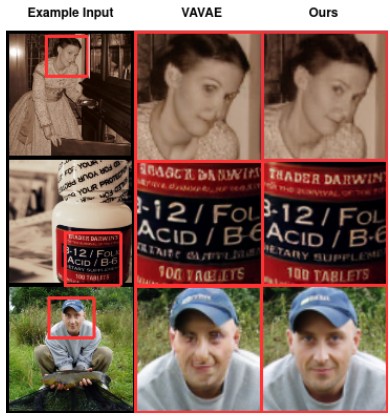

**Figure 1:** Visual comparison of reconstructions. From left to right: example original image, VAVAE reconstruction, and our approach (FA-VAE). The red color highlighted regions emphasize areas rich in textures, edges, and text. Our method better preserves high-frequency details and sharp structures, resulting in reconstructions visually closer to the input.

**Our main contributions are summarized as follows:**

- ✓ We provide a frequency-based analysis of latent tokenizers used in latent generative pipelines, showing that existing VAE variants disproportionately emphasize low-frequency components at the expense of fine detail.

- ✓ We introduce **FA-VAE**, a frequency-aware VAE framework that decouples optimization of low- and high-frequency subbands using wavelet decomposition, resulting in frequency aware expressive latent representations.

## 2 Method

### 2.1 Frequency Evaluation of Latent Embeddings

Latent tokenization models typically begin with a Variational Autoencoder (VAE) or its tokenizer variants, which learn compact embeddings $\mathbf{z} \in \mathcal{Z}$ from input data $\mathbf{x} \in \mathcal{X}$. Reconstruction quality is often measured by the pixel-level error.

$$\mathcal{L}_{\text{rec}} = \|\mathbf{x} - \hat{\mathbf{x}}\|_2^2, \quad \hat{\mathbf{x}} = D(E(\mathbf{x})),$$

where $E$ and $D$ are the encoder and decoder.

To study fidelity across frequency bands, we apply a discrete wavelet transform (DWT) with Haar filter to decompose signals into low- and high-frequency bands with one decomposition level:

$$\mathcal{W}(\mathbf{x}) = (\mathbf{x}_L, \mathbf{x}_H), \quad \mathcal{W}(\hat{\mathbf{x}}) = (\hat{\mathbf{x}}_L, \hat{\mathbf{x}}_H),$$

Then, we compute the frequency-aware reconstruction losses as follows:

$$\mathcal{L}_L = \|\mathbf{x}_L - \hat{\mathbf{x}}_L\|_2^2, \quad \mathcal{L}_H = \|\mathbf{x}_H - \hat{\mathbf{x}}_H\|_2^2.$$

Beyond pixel errors, perceptual similarity is captured by LPIPS [14], and distributional alignment by reconstruction FID (rFID) [15], computed from Inception features [16]. Together, combining frequency-aware losses $(\mathcal{L}_L, \mathcal{L}_H)$, perceptual similarity (LPIPS), and rFID provides a more complete evaluation of reconstruction fidelity and expressiveness of the learned latent embeddings.

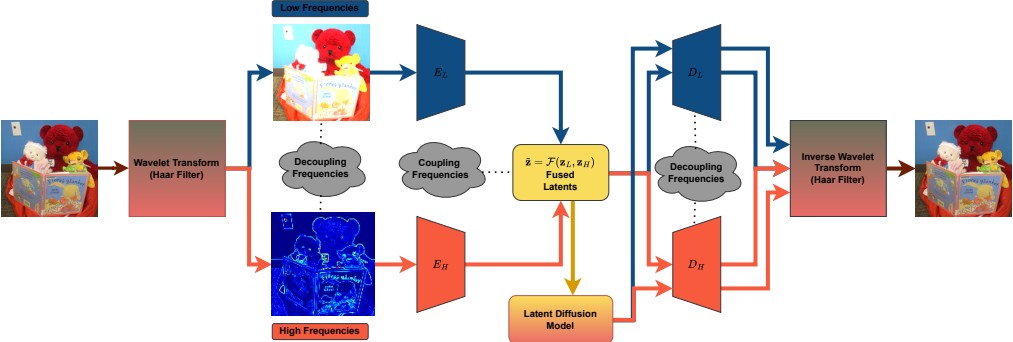

**Figure 2: FA-VAE.** The Overall pipeline of FA-VAE latent tokenizer.

## 2.2 Frequency-Aware VAE (FA-VAE)

The overview of FA-VAE is shown in Fig. 2. To enhance the fidelity of latent embeddings, we extend the standard VAE formulation by decoupling the input image into low- and high-frequency components, which are then learned independently. Given an input image $\mathbf{x} \in \mathcal{X}$, we apply a discrete wavelet transform $\mathcal{W}(\cdot)$ to obtain:

$$\mathcal{W}(\mathbf{x}) = (\mathbf{x}_L, \mathbf{x}_H),$$

where $\mathbf{x}_L$ and $\mathbf{x}_H$ denote the low- and high-frequency representations, respectively. We use the Haar filter for wavelet transformation, and apply the normalization strategy proposed in [17] to normalize both low and high frequency components. We employ separate encoder–decoder pairs $(E_L, D_L)$ and $(E_H, D_H)$ to learn frequency-specific latent embeddings:

$$\mathbf{z}_L = E_L(\mathbf{x}_L), \quad \mathbf{z}_H = E_H(\mathbf{x}_H).$$

**Low-Frequency Objective.** To learn low-frequency latent embeddings, we adopt a VA-VAE-style objective [7], incorporating a vision foundation alignment loss $\mathcal{L}_{\text{VF}}^L$ [7], adversarial regularization $\mathcal{L}_{\text{GAN}}^L$ inspired by VQGAN [18], and an additional perceptual loss $\mathcal{L}_{\text{LPIPS}}^L$ [14] to improve visual quality. The total low-frequency objective is defined as:

$$\mathcal{L}_{\text{low}} = \mathcal{L}_{\text{rec}}^L + \beta \cdot \mathcal{L}_{\text{KL}}^L + \lambda_{\text{VF}} \cdot \mathcal{L}_{\text{VF}}^L + \lambda_{\text{GAN}} \cdot \mathcal{L}_{\text{GAN}}^L + \lambda_{\text{LPIPS}} \cdot \mathcal{L}_{\text{LPIPS}}^L$$

where:

$$\mathcal{L}_{\text{rec}}^L = \mathbb{E}_{q(\mathbf{z}_L|\mathbf{x}_L)} \left[ \|\mathbf{x}_L - D_L(\mathbf{z}_L)\|_2^2 \right],$$

$$\mathcal{L}_{\text{KL}}^L = D_{\text{KL}} \left( q(\mathbf{z}_L \mid \mathbf{x}_L) \,\|\, p(\mathbf{z}_L) \right),$$

with $q(\mathbf{z}_L \mid \mathbf{x}_L)$ being the encoder's approximate posterior and $p(\mathbf{z}_L) \sim \mathcal{N}(\mathbf{0}, \mathbf{I})$, the standard gaussian prior. The vision foundation loss $\mathcal{L}_{\text{VF}}^L$ aligns the latent codes with the feature space of a pretrained foundation model (e.g., DINOv2 [19]). The adversarial loss $\mathcal{L}_{\text{GAN}}^L$ introduces a discriminator to distinguish real from reconstructed low-frequency inputs. Finally, the perceptual loss $\mathcal{L}_{\text{LPIPS}}^L$ encourages perceptual similarity between input and reconstruction using features from a pretrained Inception network [16]. All these loss components are commonly used in modern VAE frameworks.

**High-Frequency Objective.** In contrast, high-frequency components are trained without supervision from pretrained models, as they tend to be biased toward low-frequency content. We use a lightweight VAE objective focused on reconstructing fine-scale details, along with adversarial regularization:

$$\mathcal{L}_{\text{high}} = \mathcal{L}_{\text{rec}}^H + \beta \cdot \mathcal{L}_{\text{KL}}^H + \mathcal{L}_{\text{GAN}}^H,$$

$$\mathcal{L}_{\text{rec}}^H = \mathbb{E}_{q(\mathbf{z}_H|\mathbf{x}_H)} \left[ \|\mathbf{x}_H - D_H(\mathbf{z}_H)\|_1 \right],$$

$$\mathcal{L}_{\text{KL}}^H = D_{\text{KL}} \left( q(\mathbf{z}_H \mid \mathbf{x}_H) \,\|\, p(\mathbf{z}_H) \right)$$

**At inference**, both decoders reconstruct the respective frequency bands, and the final image is synthesized via inverse wavelet transform. This frequency-aware latent tokenization via FA-VAE yields more expressive embeddings by preserving details across the full spectrum of spatial frequencies.

$$\hat{\mathbf{x}}_L = D_L(\mathbf{z}_L), \quad \hat{\mathbf{x}}_H = D_H(\mathbf{z}_H), \quad \hat{\mathbf{x}} = \mathcal{W}^{-1}(\hat{\mathbf{x}}_L, \hat{\mathbf{x}}_H).$$

| Model (Tokenizer) | Tokenizer Config | Recon. Loss | Low Freq. Loss | High Freq. Loss | LPIPS | rFID |
|---|---|---|---|---|---|---|
| DC-AE | f32c32 | 0.0194 | 0.0484 | 0.0097 | 0.1580 | 0.7450 |
| DC-AE | f64c128 | 0.0207 | 0.0527 | 0.0100 | 0.1667 | 0.7623 |
| DC-AE | f128c512 | 0.0225 | 0.0581 | 0.0106 | 0.1805 | 0.7912 |
| SD-VAE | f16c16 | 0.0180 | 0.0422 | 0.0100 | 0.1743 | 0.6213 |
| KL-VAE | f16c16 | 0.0148 | 0.0326 | 0.0089 | 0.1355 | 0.5318 |
| MS-VQ-VAE | f16c32v4096 | 0.0195 | 0.0549 | 0.0076 | 0.1890 | 0.6981 |
| RQ-VAE | f32c256v16384 | 0.0416 | 0.1219 | 0.0149 | 0.2712 | 0.9095 |
| TiTok-VQ-VAE | f256x2c64v8192 | 0.0226 | 0.0561 | 0.0114 | 0.2082 | 0.7550 |
| TiTok-VQ-VAE | f256x4c64v8192 | 0.0339 | 0.0947 | 0.0137 | 0.2657 | 0.8823 |
| TiTok-VQ-VAE | f256x8c64v8192 | 0.0450 | 0.1344 | 0.0152 | 0.2949 | 0.9416 |
| TiTok-VAE | f256x1c16 | 0.0332 | 0.0923 | 0.0134 | 0.2232 | 0.8640 |
| TiTok-VAE | f256x2c16 | 0.0461 | 0.1380 | 0.0155 | 0.2682 | 0.9187 |
| TiTok-VAE | f256x4c16 | 0.0617 | 0.1961 | 0.0170 | 0.3182 | 0.9761 |
| VQ-VAE | f16c256v1024 | 0.0492 | 0.1478 | 0.0163 | 0.3064 | 0.9102 |
| VQ-VAE | f16c256v16384 | 0.0438 | 0.1262 | 0.0164 | 0.2784 | 0.8826 |
| **VA-VAE** | **f16c32** | **0.0105** | **0.0200** | **0.0074** | **0.0975** | **0.4884** |
| **FA-VAE (Ours)** | **f16c32** | **0.0044** | **0.0114** | **0.0020** | **0.0940** | **0.4156** |

**Table 1:** Quantitative comparison of latent tokenizers. Configurations: **f** = latent spatial resolution, **c** = latent dimensionality, **v** = vocabulary size (for quantized models). Lower is better.

## 3   Experiments

To evaluate the reconstruction performance of latent tokenizers in both the spatial and spectral (frequency) domains, we consider a range of widely adopted visual tokenizers commonly used in modern latent generative models. These tokenizers are based on different variants of autoencoding architectures, each aiming to learn compact and informative latent embedding representations of the input data distribution. For a comprehensive analysis, we include recent representative tokenizers based on standard Autoencoders [20], Variational Autoencoders (VAEs) [21, 6, 22, 23, 7], and Vector Quantized Autoencoders (VQ-VAEs) [23, 18, 11]. We also include our proposed FA-VAE, which explicitly incorporates frequency-awareness into the latent embedding learning process. This selection enables a fair comparison across diverse tokenization strategies with varying latent dimensionalities (c) and latent feature resolutions (f) as shown in Table 1 with diverse tokenization configurations. We use the metrics discussed in method section 2.1 for our evaluation.

Our analysis in Table 1 shows that KL-regularized VAEs with carefully designed parameterizations consistently outperform both standard VAEs and VQ-VAE baselines across all reconstruction metrics, achieving efficient performance at higher latent compression rates due to the absence of quantization artifacts. Their effectiveness is strongly tied to latent parameterization quality, as seen in recent autoregressive models [24] and VA-VAE [7], which further align latent spaces with foundation models (e.g., DINOv2 [19]) for improved perceptual quality. However, such models jointly optimize frequency components, leading to trade-offs and suboptimal preservation of low and high-frequency details (Especially high frequency details). In contrast, our FA-VAE explicitly decouples low- and high-frequency bands, enabling specialized representation learning and more precise reconstructions of both global structures and fine details. FA-VAE nearly halves the reconstruction loss of the strongest baseline while achieving the best overall performance across both low and high frequency reconstructions, demonstrating the benefits of frequency-aware modeling. The consistent gap between low- and high-frequency reconstructions across models further underscores the general difficulty of capturing fine details effectively. The other metrics like LPIPS and rFID showcases the perceptual quality of reconstructions of FA-VAE compared with other baselines.

## 4   Conclusion

In this work, we investigate frequency awareness in the latent embeddings of latent tokenizer models. We find that jointly optimizing low- and high-frequency components leads to a frequency bias favoring low-frequencies thereby degrading the reconstruction of fine-grained, high-frequency details and overall perceptual quality. To address this limitation, we introduce **FA-VAE**, a frequency-aware VAE framework that explicitly decouples low and high-frequency components using wavelet decomposition, processes them independently, and fuses them into a unified latent representation. FA-VAE achieves state-of-the-art performance across frequency-aware reconstruction metrics, demonstrating improved fidelity and perceptual quality of the learned latent embeddings.

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

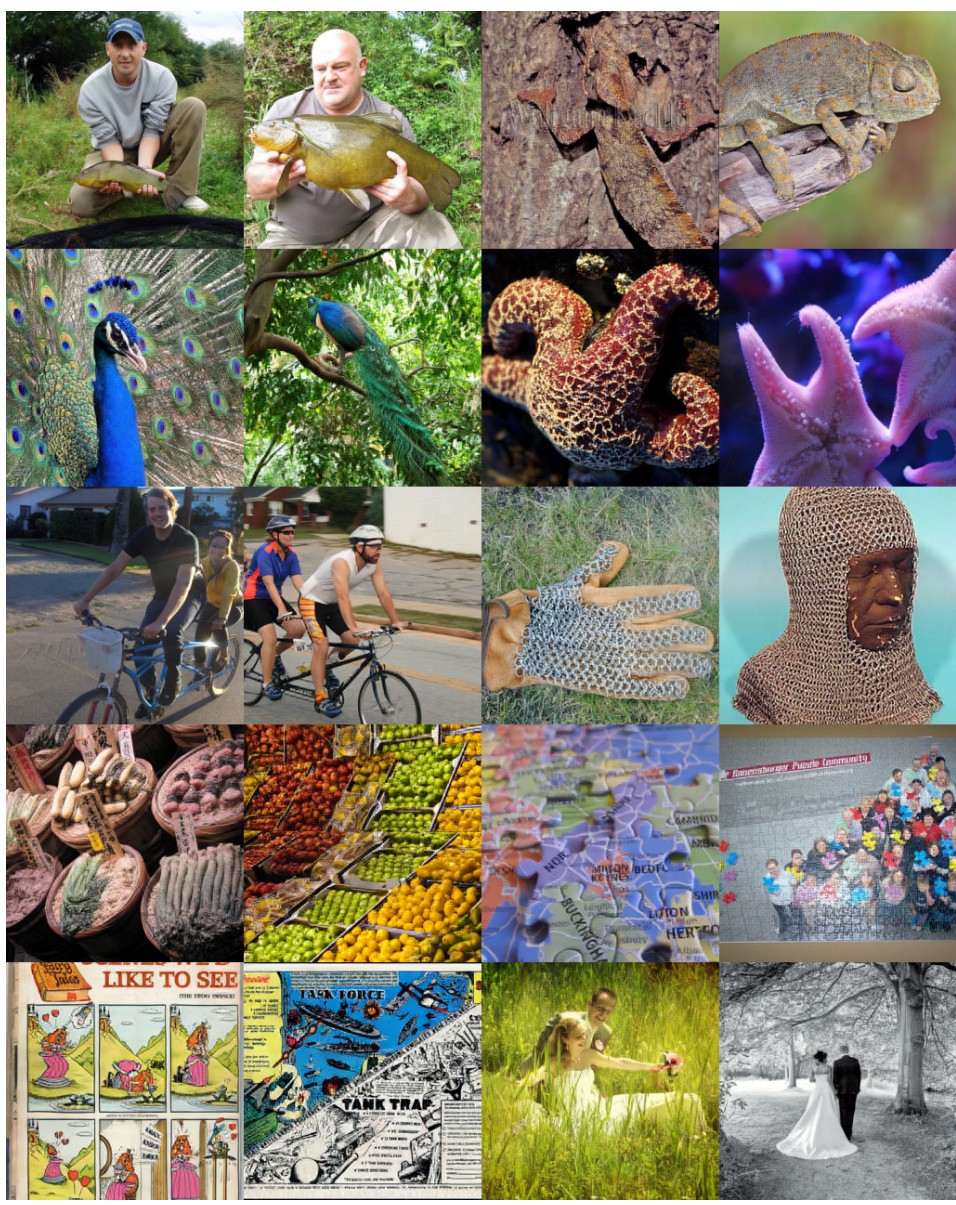

**Figure 3:** Qualitative reconstructions using **FA-VAE** on ImageNet 256×256.

## A   Architecture and Implementation Details

The architecture of our Frequency-Aware Variational Au- toencoder (FA-VAE) consists of two independent encoder- decoder pairs, designed to separately process low- and high-frequency components of the input data. The encoder- decoder pair architecture is inspired by VAVAE [7]. We further follow a similar hyperparameter setup to [21, 7] for implementing our latent reconstruction module. To support distributed training across multiple nodes, we scale the learning rate and global batch size to $1 \times 10^{-4}$ and 256, respectively, following the MAR setup [6]. We experiment with two $f16$ tokenizers for low and high frequency subbands: one trained without visual alignment VF loss for high frequencies and one with VF loss using DINOv2 [19] for low frequencies. Here, $f$ represents the latent spatial resolution factor and $d$ the latent dimensionality. Following [7], we set the VF loss hyperparameters to $m_1 = 0.5$, $m_2 = 0.25$, and $w_{\text{hyper}} = 0.1$. Figure 3 shows the qualitative reconstructions of FA-VAE.

