# OpenReview forum: "FAVAE-Effective Frequency Aware Latent Tokenizer"
_NeurIPS.cc/2025/Workshop/Reliable_ML — NeurIPS 2025 - Reliable ML Workshop_

### Official Review · Reviewer_6fZH · 2025-09-19
**FA-VAE cleanly decouples low/high frequencies and materially improves reconstruction**

**Rating:** 7
**Confidence:** 2

**Review:**

The authors diagnose a low-frequency bias in latent tokenizers and propose FA-VAE, which uses a wavelet (Haar) split to learn separate low- vs high-frequency latents with tailored objectives (low-band: VF alignment + GAN + perceptual; high-band: lightweight VAE + GAN), followed by fusion via inverse wavelet. The paper reports strong reconstruction/perceptual gains across many tokenizer baselines and shows qualitative improvements on ImageNet-256 examples.

Pros: The paper states a clear problem and introduces a simple, modular fix that integrates with latent pipelines. The experiments show sizable wins over VA-VAE and VQ-based tokenizers: for f16c32, FA-VAE achieves reconstruction 0.0044 vs 0.0105 (VA-VAE), low-freq loss 0.0114 vs 0.0200, high-freq loss 0.0020 vs 0.0074, LPIPS 0.0940 vs 0.0975, and rFID 0.4156 vs 0.4884 (lower is better).

Cons: The paper evaluates only reconstruction-side metrics and does not test a downstream generator (e.g., diffusion/AR) trained on FA-VAE latents; therefore, end-to-end benefits for generation remain unproven. The evaluation appears limited to ImageNet-style data, and the manuscript lacks ablations on wavelet choice/levels, alignment loss, and L1 vs L2 for the high-band.

Suggestions: The authors should: (1) plug FA-VAE into at least one latent diffusion and one autoregressive model and report generation metrics; (2) add multi-dataset tests (e.g., FFHQ/LSUN/LAION subset or a medical imaging set) and multi-seed statistics; (3) ablate wavelet family and depth, loss components, and band-balancing weights;